# Microwave Dielectric Response of Bovine Milk as Pregnancy Detection Tool in Dairy Cows

**DOI:** 10.3390/s24092742

**Published:** 2024-04-25

**Authors:** Cindy Galindo, Guy Levy, Yuri Feldman, Zvi Roth, Jonathan Shalev, Chen Raz, Edo Mor, Nurit Argov-Argaman

**Affiliations:** 1Institute of Applied Physics, The Hebrew University of Jerusalem, Jerusalem 9190401, Israel; cindy.galindohur@mail.huji.ac.il (C.G.); guy.levy4@mail.huji.ac.il (G.L.); yurif@mail.huji.ac.il (Y.F.); 2Animal Science Department, The Robert H. Smith Faculty of Agriculture, Food, and Environment, The Hebrew University of Jerusalem, Rehovot 7610001, Israel; 3The Racah Institute of Physics, The Hebrew University of Jerusalem, Jerusalem 9190401, Israel

**Keywords:** microwave dielectric spectroscopy, bovine milk, water, non-invasive pregnancy detection

## Abstract

The most reliable methods for pregnancy diagnosis in dairy herds include rectal palpation, ultrasound examination, and evaluation of plasma progesterone concentrations. However, these methods are expensive, labor-intensive, and invasive. Thus, there is a need to develop a practical, non-invasive, cost-effective method that can be implemented on the farm to detect pregnancy. This study suggests employing microwave dielectric spectroscopy (MDS, 0.5–40 GHz) as a method to evaluate reproduction events in dairy cows. The approach involves the integration of MDS data with information on milk solids to detect pregnancy and identify early embryonic loss in dairy cows. To test the ability to predict pregnancy according to these measurements, milk samples were collected from (i) pregnant and non-pregnant randomly selected cows, (ii) weekly from selected cows (*n* = 12) before insemination until a positive pregnancy test, and (iii) daily from selected cows (*n* = 10) prior to insemination until a positive pregnancy test. The results indicated that the dielectric strength of Δ*ε* and the relaxation time, *τ*, exhibited reduced variability in the case of a positive pregnancy diagnosis. Using principal component analysis (PCA), a clear distinction between pregnancy and nonpregnancy status was observed, with improved differentiation upon a higher sampling frequency. Additionally, a neural network machine learning technique was employed to develop a prediction algorithm with an accuracy of 73%. These findings demonstrate that MDS can be used to detect changes in milk upon pregnancy. The developed machine learning provides a broad classification that could be further enhanced with additional data.

## 1. Introduction

Milk is a complex biosystem that consists of 87% water [1]. Water is considered an integral component of biomolecular systems, moreover, the properties of water depend on the structure and dielectric character of the materials in its proximity [2,3]. In biological systems, water is typically categorized into hydration (bound) water, found near biomolecular surfaces, and bulk water [4]. The dynamic properties of bulk and bound water are strikingly different [5]. Accordingly, it has been suggested that variations in solid compositions or macromolecular structures in milk could be reflected by alterations in the dynamics and structure of bulk water [6,7]. Microwave dielectric spectroscopy (MDS) is a powerful tool for studying water dynamics in complex materials [8]. For instance, using MDS on contaminated bovine milk enables evaluation of the milk quality and identification of whether the cow is unhealthy (i.e., having mastitis) or healthy (clean) [7].

The specific architecture of a water molecule can be geometrically represented by an almost regular tetrahedron with the heavy oxygen in its center, the two light hydrogens at two of the vertices, and the lone pair electrons occupying both other vertices (Figure 1) [9]. 

The electric charges in this structure are not evenly spread, leading to two important features that are primarily investigated using relaxation techniques such as MDS [9]: (a)Water molecules possess a permanent dipole moment (μ_d_ = 1.84 ± 0.02 D)(b)Water molecules can form hydrogen bonds between different molecules.

The permanent dipole moment leads to the possibility of the water molecule coupling to external applied electric fields and can be used as a naturally present marker of the molecular orientation [9]. The second property leads to interactions between the positive hydrogen atom site and a lone pair electron site of another molecule, forming a dimer. The most stable configuration of a hydrogen bond corresponds to a linear H–O…H structure. Since water molecules are each able to form four hydrogen bonds, with a symmetric distribution of two proton-donating and two proton-accepting sites, a three-dimensional hydrogen bond network (water cluster) develops in the condensed phases [9,10,11].

When a microwave alternating electric field interacts with a complex material such as milk, where water is the major component, the charge distributions and polarization of the system are changed. The induced motions and reorientations result in the dissipation of energy. In the MDS technique, the response of the particular material to the applied field can be characterized by measuring the complex dielectric permittivity [12].

The complex dielectric permittivity of pure water exhibits a peak of dielectric relaxation at 25 °C in the microwave frequency band (0.5 to 40 GHz) (Figure 2). The dielectric dispersion of water in milk is very well described by Equation (1).
(1)$$\varepsilon \left(\omega \right)=\varepsilon_{\infty }+\frac{\Delta \varepsilon }{1+{\left(i\omega \tau \right)}^{\alpha }}+\frac{\sigma_{dc}}{i\omega \varepsilon_{0}}\;+A{\left(i\omega \right)}^{n-1}$$

Here, $\varepsilon_{\infty }+\frac{\Delta \varepsilon }{1+{\left(i\omega \tau \right)}^{\alpha }}$ is the Cole–Cole function (CC) [7] and is associated with the relaxation of bulk water in milk. The CC function includes the time necessary for the material to reach equilibrium, also known as τ; the relaxation time (i.e., τ= $\frac{1}{2\pi f_{max}}$); Δε, the dielectric strength; and α, referred to as a measure of symmetrical broadening of the dielectric relaxation peak. $\varepsilon_{\infty }$ is the high-frequency limit of the dielectric permittivity, and $\varepsilon_{0}\;$ ≅ 8.85 × 10^−12^ F/m is the vacuum permittivity. In the second term, ($\frac{\sigma_{dc}}{i\omega \varepsilon_{0}}$) represents the *dc* conductivity generated by the mobility of ions. The last term is the left Jonscher function, A${\left(i\omega \right)}^{n-1}$, which represents the tail of the lower frequency process.

Based on infrared examination, it has been reported that the pregnancy state in cows induces changes in bovine milk such as altered milk yield, fat, and protein content [13]. As a biological fluid that may reflect reproductive events in dairy animals, milk is an easily accessible fluid that contains key information about a cow’s reproductive status [13,14]. 

Reproductive performance plays a pivotal role in the profitability of a dairy herd. Inefficient reproductive management, in particular, false-negative or false-positive pregnancy diagnosis, might cause substantial economic loss depending on the availability of replacement heifers, average herd lactation, milk production level, and calving interval [15]. Early identification of nonpregnant cows after insemination is therefore crucial and might shorten the time to the next insemination, thereby shortening the time that the cow is not pregnant and improving the farms’ profitability. Furthermore, it is estimated that about 20% of embryos are lost at an early stage of pregnancy [16]; therefore, timely identification of the pregnancy status is highly important for the proper reproductive planning of the herd and to maximize the production efficiency of the herd. 

To detect pregnancy in dairy cows, a physical examination using transrectal palpation, conducted at 42 days after artificial insemination (AI), is commonly used in Israel. An earlier diagnosis can be performed using ultrasound. Alternatively, evaluating the progesterone concentration in the plasma or milk is an additional approach used for pregnancy diagnosis [17]. Progesterone is the main steroid hormone in bovine that is secreted by the corpus luteum and placenta, with a high concentration in plasma throughout the pregnancy. Nevertheless, this method is expensive and is labor-intensive and consequently rarely used. Therefore, there is a need to develop non-invasive, high-throughput methods on farms to enable the detection of reproductive events, particularly pregnancy.

Previously, we reported that the milk fat globule (MFG) size is associated with progesterone concentrations [14]. This study provided insight into the association between the milk constituents and the animal reproductive status. More specifically, using an in vitro model, it was shown that progesterone, a major reproductive hormone, affects the MFG size. Moreover, changes in progesterone concentrations in plasma and milk during the estrous cycle and the transition between non-pregnant, pregnant, and postpartum states may be attributed to the change in MFG size recorded during the estrous cycle of dairy cows. 

Here, we hypothesize that the changes caused in milk fat composition and structure induced by changes in progesterone levels [14] due to reproductive events (i.e., pregnancy, early embryonic loss, and abortion) have a significant effect on the balance of bound/bulk water and can be identified using MDS. 

The primary aim of this study was to observe the alterations in the Cole–Cole (CC) fitting parameters of the complex dielectric permittivity spectra of raw bovine milk caused by reproductive events to explore the potential for developing a non-invasive approach for early pregnancy prediction. The microwave dielectric fitting parameters have been statistically analyzed using principal component analysis (PCA). Additionally, we suggest the use of an artificial neural network (ANN) that could predict pregnancy in a faster period than traditional methods.

## 2. Materials and Methods

### 2.1. Materials: Milk Provision

This work has been divided into three different sampling protocols: the first two protocols involved weekly and daily sampling from individual cows. While the weekly measurements offered insights into the dependence of the milk composition and dielectric properties on the pregnancy status, potential trends and patterns became more evident with daily sampling due to the higher frequency of data collection. The third sampling protocol was the collection of milk samples from randomly selected pregnant cows at various stages of pregnancy (first, second, and third trimester) and non-pregnant cows. This protocol was conducted to generate a larger number of samples required for the machine learning phase. 

#### 2.1.1. Weekly Samples

Raw milk samples were collected from Israeli Holstein lactating cows (*n* = 12) at the Beit Dagan Farm, Israel. Holstein cows in their first to fifth lactation were used. Cows were kept in an open field and fed ad libitum a total mixed ration containing 1.75 Mcal of NE_L_/Kg dry matter and 17% protein, which is the standard ration in Israel. Cows were milked three times daily. The samples were taken at seven-day intervals at the first milking (3:00 a.m.), divided into aliquots, and transported at 4 °C to the Center of Electromagnetic Research and Characterization (CERC; The Hebrew University of Jerusalem) for MDS measurements and milk solids measurements. The study lasted 8 months; for each examined cow, the sampling started while the cow was empty (i.e., not pregnant), then was performed before and after insemination until a positive pregnancy test. Once a cow was confirmed as pregnant, it was excluded from the study and replaced by a new nonpregnant cow. The concentration of milk solids (fat, protein, and lactose) was measured using the infrared methodology (Lactoscan; Miltkotronic, Nova Zagora, Bulgaria). Data on the somatic cell count (SCC) and days in milk were recorded by the online system (Afilab, ZHM Afikim, Israel) and the herd management software, respectively. The detailed data can be found in Appendix A).

#### 2.1.2. Daily On-Farm Measurements

Raw milk samples (10 mL/per sample) from 10 cows were collected daily from the Volcani Center dairy farm (Beit Dagan, Israel). Dielectric measurements were performed on site in a portable laboratory to avoid long handling and transportation times. Milk was collected using an automated sampling device, which collects milk throughout the milking session. Milk samples were stored on ice until milk from all the experimental cows was collected. Before analysis, the samples were allowed to reach a room temperature of 25 °C. Cows were selected before insemination according to their postpartum reproductive status and whether they exhibited standing estrus according to their pedometric activity (using NOA, The Israeli Dairy Herd Management Software, developed by the Israel Cattle Breeders Association-ICBA since 2000). Cows were sampled until a positive pregnancy test determined by palpation by the herd’s veterinarian, typically 42 days post-insemination. Overall, the study lasted for 65 days and was performed from January to March to avoid summer periods when the cows undergo heat stress. The concentration of milk solids (fat, protein, and lactose) was measured as detailed above. The activity peak (pedometric activity) and days in milk were recorded on the farm on the day of sampling. The detailed data can be found in Appendix A). 

#### 2.1.3. Measurements on Randomly Selected Pregnant and Non-Pregnant Cows

Raw milk samples were collected every week from randomly selected pregnant and non-pregnant cows at the Beit Dagan Farm in Israel. The samples were obtained from the first milking, which took place at 3:00 a.m., to ensure consistency. Subsequently, the samples were promptly divided into aliquots and transported at a temperature of 4 °C to the Faculty of Agriculture, Hebrew University of Jerusalem for milk solids measurements, as well as to the Center of Electromagnetic Research and Characterization (CERC) at The Hebrew University of Jerusalem to undergo MDS measurements. 

The study was conducted for three months, specifically from December to March, to avoid the summer season when cows may experience heat stress. During this period, a total of 117 milk samples were collected. Out of these, 50 samples corresponded to pregnant cows, while the remaining 67 samples were from non-pregnant cows. The concentration of milk solids (fat, protein, and lactose) was measured as described above. The somatic cell count (SCC) in milk was recorded as described above. The detailed data can be found in Appendix A). 

### 2.2. Methods: Microwave Dielectric Spectroscopy Measurements

Dielectric measurements were made using a PNA Network Analyzer (N523B PNA-L) in the frequency range from 500 MHz to 40 GHz. The PNA was connected to a Keysight N1501A dielectric kit, including a performance coaxial probe. The system was calibrated using three standards: air, short, and double-distilled water at 25 °C. Precision in the calibration was achieved using an N4694-60001 two-port microwave electronic calibration module (Ecal). A special stand for the performance probe was designed and combined with the sample cell for the milk samples (total volume—7.8 mL). For dielectric measurements in the CERC, the temperature was controlled using a thermal jacket attached to a Julabo C-41 oil-based heat circulatory system. The cell was held at 25 °C with temperature fluctuations less than 0.5 °C. The whole experimental setup is presented in Figure 3.

Regarding the dielectric measurements on the Beit Dagan Farm, the whole measuring station was placed in an air-conditioned room at 25 ± 1 °C. The sample temperature was controlled by placing a thermometer inside the sample. Each sample was measured at least three times; before each measurement, the coaxial probe was removed and cleaned with double-distilled water to avoid the formation of fat layers. Moreover, the sample was mixed using clean pipettes. The real and imaginary parts of the complex dielectric permittivity were obtained using the Keysight Materials Measurement Suite 2018 [18] with an accuracy of Δ*ε*′/*ε* = 0.05, Δ*ε*″/*ε* = 0.05. 

### 2.3. Milk Composition Data

The composition of milk solids, including fat, protein, and lactose percentages, was determined on the same day as milk collection using a near-infrared scanning device (Miltkotronic, Nova Zagora, Bulgaria) according to our previous study [19]. Briefly, milk samples were stained with fluorescence dye (Nile red, Sigma Aldrich, Rehovot, Israel), and lipid droplets were visualized under a fluorescence microscope. Images were analyzed using ImageJ to determine the milk fat globule average diameter in each milk sample. 

### 2.4. Microwave Dielectric Spectra Fitting

The complex dielectric spectra were fitted using an in-house MATLAB-developed program, DATAMA [20]. The standard deviation (SD) of each fitting parameter (Equation (1)) from at least three measurements for each sample was calculated.

### 2.5. Machine Learning

The artificial neural network (ANN) algorithm utilized in this study was the self-normalizing network (SNN) algorithm [21]. The SNN method has been shown to outperform other fully connected networks for a variety of classification and regression tasks concerning numeric data [21]. This class of models provides optimal propagation of activations that are close to zero mean and unit variance across many layers. This is achieved using a special activation function known as the scaled exponential linear unit (SELU) and a special type of dropout named Alpha Dropout [21]. The SNN architecture consisted of eight scaled exponential linear snits (SELUs), each with a width of 50 neurons. To mitigate overfitting, we applied a dropout probability of 0.01 to each layer.

The dielectric and physiological measurements were classified as “PREGNANT” and “NONPREGNANT” to construct a training algorithm with a reasonably large data set. For that, we used the data obtained from the daily and weekly samples as well as data from milk collected from random cows. No individual cows were considered. The total number of points used to build the algorithm was equal for pregnant and nonpregnant cows. The measurements were selected from a pool of 168 measurements of milk from pregnant cows and 642 measurements of milk from nonpregnant cows. As the datasets are biased toward nonpregnant features, the nonpregnant features were randomly down-sampled to match the number of pregnant features in each set. In our case, the input layer was composed of four dielectric parameters.

After each update during training, the model underwent evaluation on both the training dataset and a separate validation dataset using a hold-out technique. This technique involved splitting the original dataset into two sets: a “train” set and a “test” set. We employed a 90/10 split, where 90% of the data was allocated to the training set and the remaining 10% to the test set. The model was trained on the training set, while the test set served as a benchmark to assess the model’s performance on unseen data. Learning curves depicting the measured performance were generated to illustrate the model’s progress. For our experimental setup, we preprocessed the dataset comprising 302 train points and 30 test points. The test points were randomly selected, with 15 true (pregnant) and 15 false (non-pregnant) labels, ensuring an unbiased evaluation. Each point was scaled to an order of 1.

The model training was conducted using the ADAM optimizer with specific hyperparameters including a Beta 1 = 0.9, Beta 2 = 0.999, and stability parameter Epsilon = 0.00001. The training process consisted of 20 epochs.

To evaluate the model, we utilized the cross-entropy (binary) loss function [22]. The entire analysis was conducted in Mathematica version 13.0. 

### 2.6. Statistics

The statistical analysis includes monitoring the variability in dielectric parameters, generating Pearson correlation matrices, and conducting principal component analysis (PCA). All statistical analyses were performed at the Center of Electromagnetic Research and Characterization (CERC) using OriginPro software, version 2022.

## 3. Results and Discussion

### 3.1. The Dielectric Spectrum of Water and Milk

Milk is a complex colloidal system that is primarily composed of water (~87%), alongside other constituents such as lactose, fat, proteins, and inorganic components. The mean values of certain solid components in milk from the three studied groups were determined with their respective standard deviations: fat (3.37 ± 0.54%), protein (3.36 ± 0.09%), and lactose (4.74 ± 0.11%). Remarkably, this data closely aligns with the values reported in the literature [1]. The significant portion occupied by water is of particular interest. Water is considered an integral component of biomolecular systems [23]. It is involved in determining the macromolecules’ functional and structural properties and their interactions [4]. Therefore, variations in milk solids are expected to change the properties of the water fraction in milk.

Compared with double-distilled water, the interactions between the different components of milk and water induce changes in the dielectric spectrum, such as a smaller static permittivity, a lower characteristic relaxation frequency, the broadening of the relaxation peak, and DC conductivity (Figure 2). Accordingly, a change in the static permittivity, broadening, and position of the main dispersion peak is related to the state of bulk water in the system. 

### 3.2. MDS Is Sensitive to the Changes in Bovine Milk Caused by Pregnancy

The CC fitting parameters Δ*ε*, *τ*, *α*, and *σ_dc_* (Equation (1)) were used to track the changes in the dielectric response of bulk water in milk before and after insemination. All of the fitting parameters ($\Delta \varepsilon ,\sigma_{dc}\alpha ,\tau$) had a significant decrease in variability after the first two weeks of successful insemination (i.e., an insemination followed by a positive pregnancy diagnosis). Figure 4 depicts the transition to pregnancy status in terms of the fitting parameters in five cows confirmed as pregnant. A significant reduction in the variability in CC data can be observed during the transition, divided into three stages, (1) beginning with a nonpregnancy condition, (2) progressing to 15 days of pregnancy, and then (3) continuing to the 16th day. The first state had 66 nonpregnancy points, the second state had 8, and the final state had 14 points. All points in each state were arranged in a single column.

The standard deviation for Δ*ε* was 0.3; for *τ* it was 0.6 (ps); for *α* it was 0.002; and for *σ* it was 0.01 (S/m).

On the other side, confirmed nonpregnant cows (*n* = 7) (Figure 5) did not exhibit such a reduction in variability. 139 nonpregnancy points were separated into two groups: before and after insemination. The first group received 70 nonpregnancy points, while the second received 69. The separation was determined when the first artificial insemination failed. In Figure 4 and Figure 5, the most noteworthy difference between nonpregnant and pregnant cows is in Δ*ε*, where there is a 44% drop in variability compared to a 3.26% drop in nonpregnant cows.

#### 3.2.1. Pearson Correlation Matrix

The correlation matrix was calculated and includes the fitting parameters of complex dielectric spectra with the composition of milk solids (Table 1).

Accordingly, it was observed that in the seven pregnant cows (five pregnant cows from the weekly experiment and two pregnant cows from the daily experiments), the fitting parameter Δε has a moderate negative correlation with the fat percentage in milk; the low *p*-value indicates that this correlation is statistically significant. The negative correlation suggests that a rise in fat percentage results in a reduction in Δε. All of the other parameters have a negligible or weak correlation. For the correlation of dielectric parameters and weather, only daily measurements were employed.

A moderate negative correlation between fat and dielectric strength (Δ*ε*) indicates that fat may play a significant role in influencing the sensitivity of microwave dielectric spectroscopy to variations in milk composition related to pregnancy status. Bovine milk’s fat component primarily consist of triacylglycerols (98%), along with some minor constituents such as diacylglycerols, monoacylglycerols, free fatty acids, phospholipids, and cholesterol [24]. Furthermore, triacylglycerols are structured as oil-in-water emulsions, commonly known as fat globules, which are surrounded by a unique complex layer called the milk fat globule membrane (MFGM) [24]. The MFGM contains an outer phospholipid bilayer where the most abundant phospholipids are phosphatidylcholine, phosphatidylethanolamine, and sphingomyelin [24]. Phospholipids and sphingolipids are renowned for their amphiphilic nature, the polar head groups contain phosphates and glycerol molecules capable of establishing hydrophilic interactions with surrounding water, thus forming hydration shells [25].

Consequently, any increase or decrease in the number of milk fat globules will alter the equilibrium between bulk and bound water within the entire system. Specifically, if the number of water molecules migrating to the hydration shells of milk fat globules increases due to an increase in their quantity or surface area, the number of water molecules in the bulk, which can respond to an external applied electric field in the microwave frequency range, will decrease. This reduction in bulk water molecules that can respond leads to a decrease in dielectric strength [9].

#### 3.2.2. Evaluation of CC Fitting Parameters Using Principal Component Analysis (PCA)

In order to achieve better accuracy in pregnancy detection, it was necessary to use other visualization techniques. Figure 6 and Figure 7 display the transformed data obtained from a single cow by projecting the complete set of CC fitting parameters (Δ*ε*, *τ*, *α*, and *σ*) into a reduced dimensional space using the first two principal components (PCs) derived from principal component analysis (PCA). PCA enables the representation of multiple interrelated variables through a smaller set of variables. Additionally, this reduced set of variables captures a significant portion of the variability present in the original data [26].

We used PCA to study the clustering of samples collected from pregnant and non-pregnant cows in the weekly and daily measurements. Using PCA, it was possible to identify three out of five pregnant cows in the weekly experiments. Regarding daily experiments, which allow better detection of changes due to greater sampling frequency, two out of two pregnant cows from total of ten cows were identified. In Figure 6, cows A & B represent two pregnant cows that belong to the weekly and daily experiments, respectively. In addition, in the case of daily measurements, the first 15 days of pregnancy have been labeled with a number. By detecting a clustering and a shift between the states in these cows’ biplots, it was feasible to differentiate between pregnancy and nonpregnancy.

Interestingly, the variability reduction presented in the previous section was also clear in the PCA for cow B (Figure 6), which contains the daily measurement results. The points in the biplot seem to form a smaller cluster after the 15th day of pregnancy (blue points). Moreover, the PCA plots of nonpregnant cows show a random distribution of points before and after insemination (Figure 7). Remarkably, in the case of confirmed pregnant cows (Figure 6), a clear distinction can be made between the nonpregnancy and the insemination states. Such clusterization is not clear in the case of nonpregnant cows (Figure 7).

#### 3.2.3. Artificial Neural Network (ANN). A Machine Learning Approach for Early Pregnancy Detection in Cows

The above-mentioned results confirmed that the changes in the balance between bound and bulk water in milk are affected by the reproductive state and have the potential to distinguish between pregnant and non-pregnant cows and be utilized for pregnancy detection. We were able to identify such changes using traditional statistics tools that are limited by human efforts. Moreover, these approaches require sequential measurements and are dependent on frequent measurements throughout a prolonged period to receive a change in the clustering of data points. Thus, a machine-learning approach known as artificial neural networks (ANNs) was used to distinguish between the two states more precisely. One of the ANN outputs is a plot called the learning curve, which displays the performance plot over time. Learning curves are a standard diagnostic tool in machine learning for algorithms that learn progressively from a training dataset.

Figure 8a represents the loss plot. The loss plot indicates the difference between prediction and truth. Hence, a machine learning algorithm’s purpose is to reduce the loss. In a good model, training and validation finally converge.

Processing the data of the present study revealed that the training data is lower than the validation data, suggesting that the information between training and validation is unrepresentative. This implies that there is slightly different behavior in training and a small overfit in the validation data; further, the validation error is greater than the training error. This phenomenon is expected when the model has more capacity than is necessary for the problem, resulting in too much flexibility. It can also happen if the model is trained for an extended period. The model is somewhat overfitting in this situation, indicating that it is not sufficiently generalized.

In this study, validation and training data did not converge into one. However, considering the small data pool used to build this algorithm, these results demonstrate that a more extended model can be developed using our four dielectric parameters and the milk composition data provided by the farm. Additionally, the accuracy plot (Figure 8b) helps to determine the quality of a model during the training. High accuracy is reached when the number of epochs increases (the number of times data is given to the neural network). The results show a 73% accuracy.

## 4. Conclusions

The present study aims to detect changes in the reproductive state of dairy cows after insemination in a high-throughput, non-invasive automated system. This approach was developed based on previous findings showing a change in the MFG size during the estrous cycle in vivo [14] and based on previous studies that showed a positive correlation between MFG size and milk fat content [27].

Moreover, in vitro, it was demonstrated that the progesterone concentration plays a central role in the change in MFG size [14]. Since these changes in the structural composition of milk fat may affect the water properties of milk, we decided to study how events that change the progesterone concentration (pregnancy, early embryonic loss) are manifested in milk water properties. Furthermore, we integrated this information into a process that will enable automated detection of the cows’ reproductive events.

The microwave dielectric measurements of milk confirmed that MDS is a potential method for early pregnancy detection in cows. The sensitivity is based on the reduction in the variability in the CC parameter: dielectric strength, Δ*ε*. After daily measurements, a reduction in variability of 44% was observed during the first 16 days of pregnancy. Such a reduction in variability was not evident in the case of nonpregnant cows. Using PCA, a clear distinction between pregnancy and nonpregnancy status was seen in the case of successful inseminations. However, the clustering was not apparent in the case of nonpregnant cows. Finally, using a machine learning method known as artificial neural networks, we demonstrated that the classification of pregnancy can be based on Δ*ε*, *τ*, *α*, and *σ_dc_* dielectric parameters. The system expressed 73% accuracy in predicting pregnancy. Nevertheless, additional data is required to generalize a more accurate model. Data on physiological parameters of milk such as solids concentrations, and especially milk fat concentrations, can contribute to the accuracy of the system, as they are associated with MFG size. MFG size was previously shown to be affected by progesterone concentrations which is the major hormone that changes due to reproductive events; moreover, inclusion of the milk fat concentration can contribute to the prediction system. Unlike laboratory progesterone tests, we suggest consequent sampling of milk and using dielectric properties which might provide us information about pregnancy and early embryonic loss. More frequent data collection can also enhance the accuracy of the model, and this can be easily achieved in commercial dairy farms since, in most of the modern farms in Israel, milking occurs three times daily. Thereby, a 99% accuracy may be achieved in training the machine learning algorithm. The method that has been developed in the current project is remarka-ble because it can be used on any farm without the need for prior information about the local cows. 

## Figures and Tables

**Figure 1 sensors-24-02742-f001:**
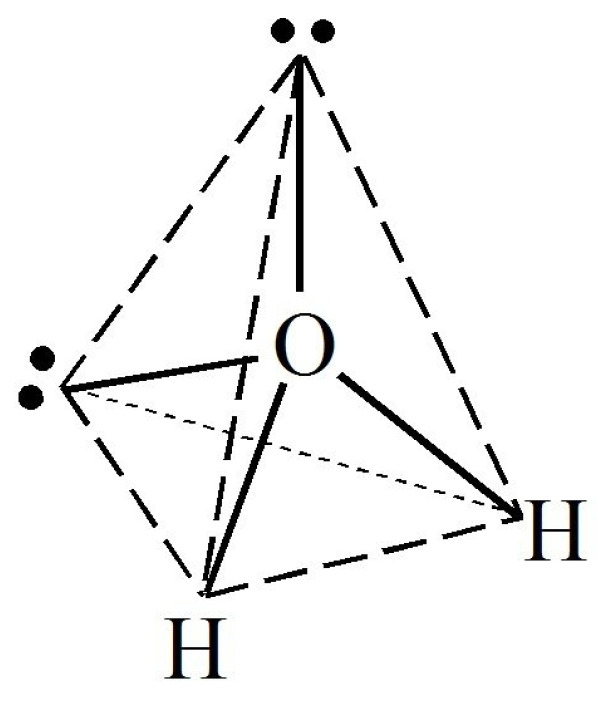
Tetrahedral structure of a water molecule.

**Figure 2 sensors-24-02742-f002:**
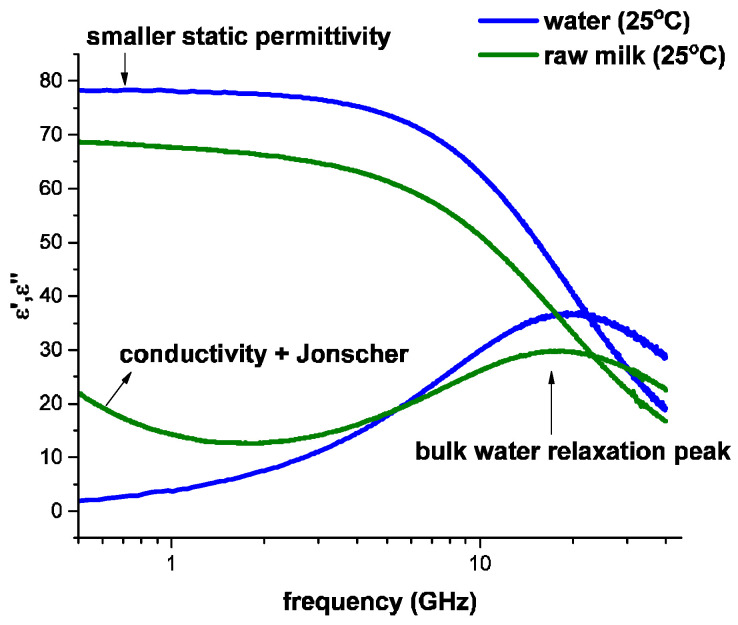
Microwave dielectric spectra of double-distilled water compared to raw milk at 25 °C and frequencies ranging from 0.5 to 40 GHz. Milk is characterized by a lower static permittivity (real (*ε*′) part). In the imaginary part (*ε*″) of the milk spectrum, the conductivity tail at lower frequencies and the broadening of the main dispersion peak can be observed.

**Figure 3 sensors-24-02742-f003:**
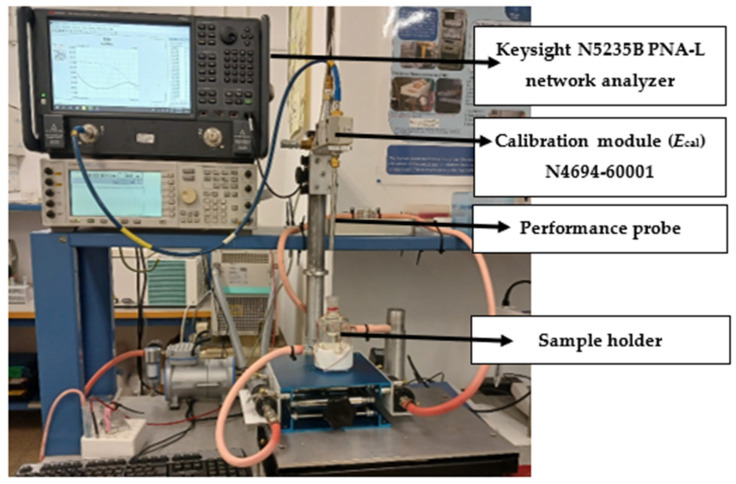
Experimental workstation for microwave dielectric measurements.

**Figure 4 sensors-24-02742-f004:**
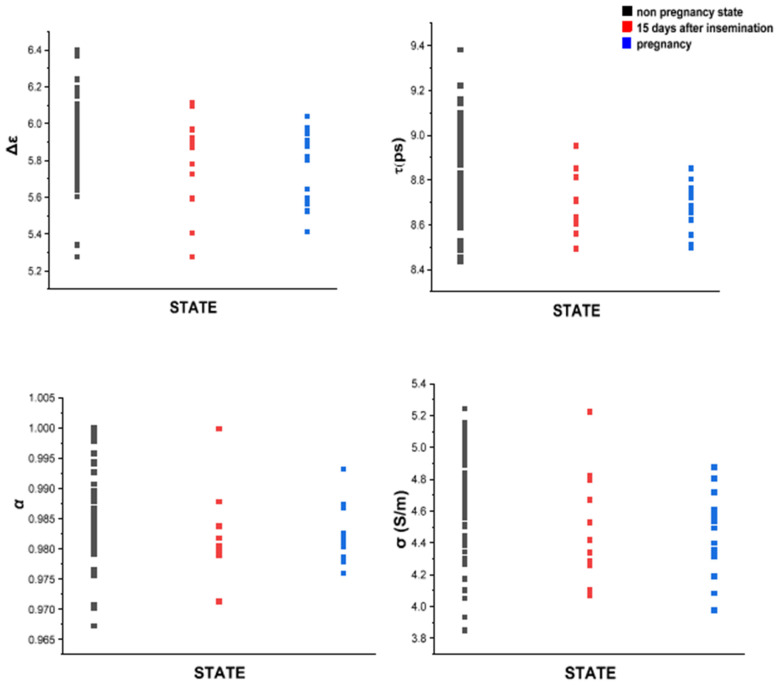
CC parameters of milk from confirmed pregnant cows. The variability in the fitting parameters significantly reduces when artificial insemination is successful. A more substantial reduction in variability is observed after the 16th day of pregnancy.

**Figure 5 sensors-24-02742-f005:**
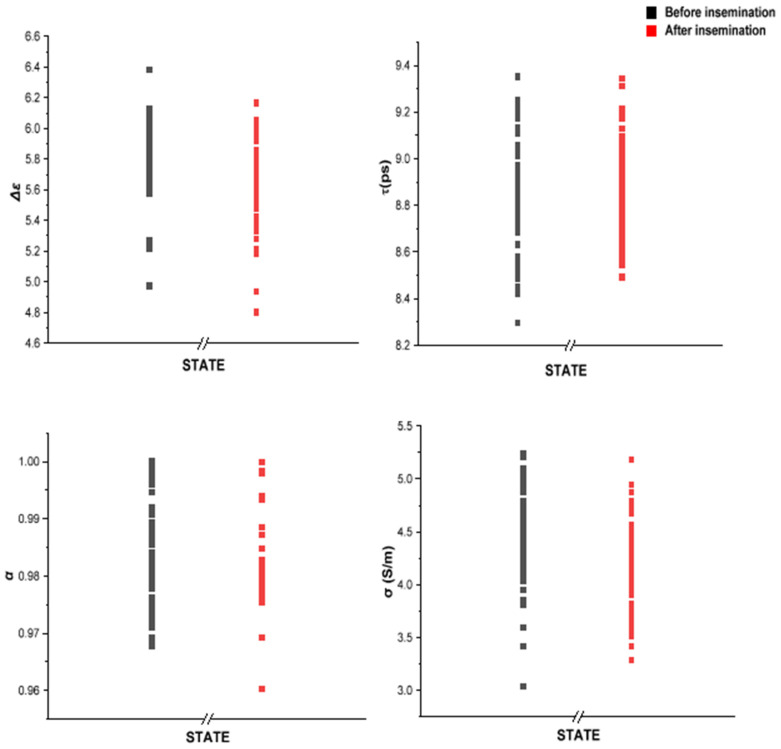
CC parameters of milk from confirmed nonpregnant cows. The variability in the fitting parameters has no significant change before and after artificial insemination. The post-insemination period after insemination is calculated starting from the exact day of insemination.

**Figure 6 sensors-24-02742-f006:**
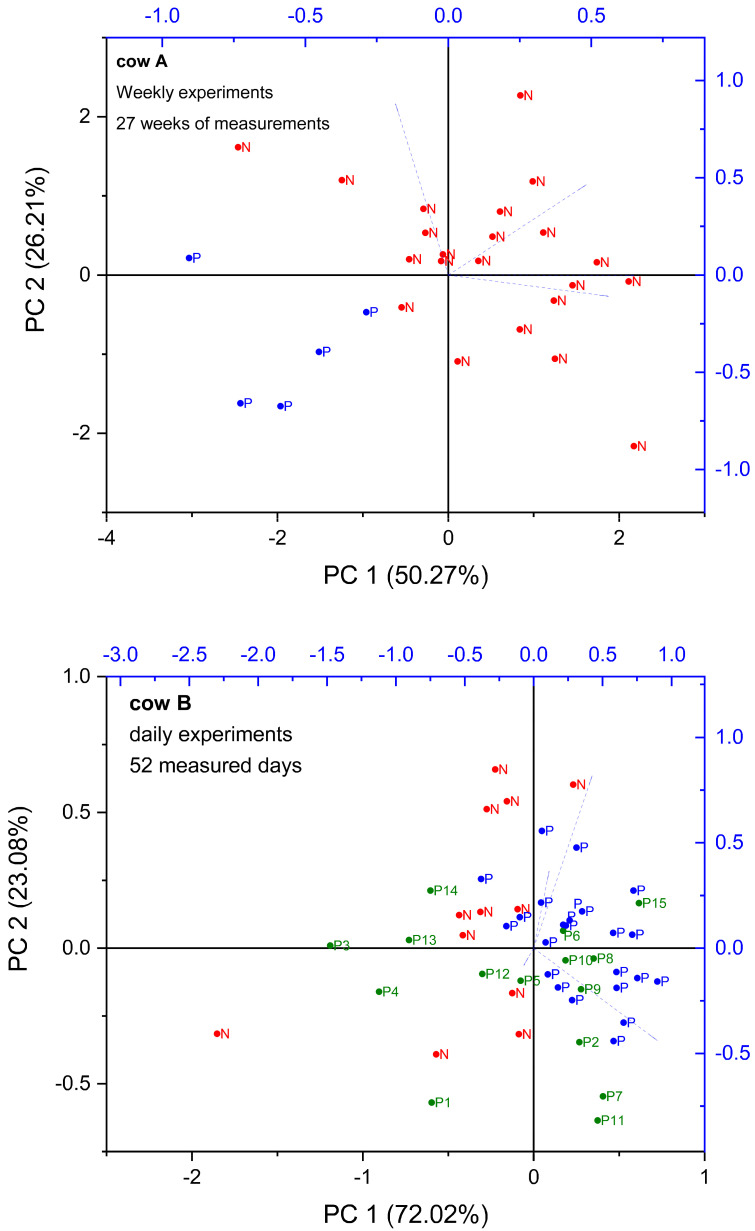
PCA biplot of the data corresponding to confirmed pregnant cows. The nonpregnant state has been marked as red N and pregnant as blue P. For cow B, the first 15 days of pregnancy have been labeled with the corresponding number in green. Cow A was measured weekly for 27 weeks, and cow B was measured daily for 52 days.

**Figure 7 sensors-24-02742-f007:**
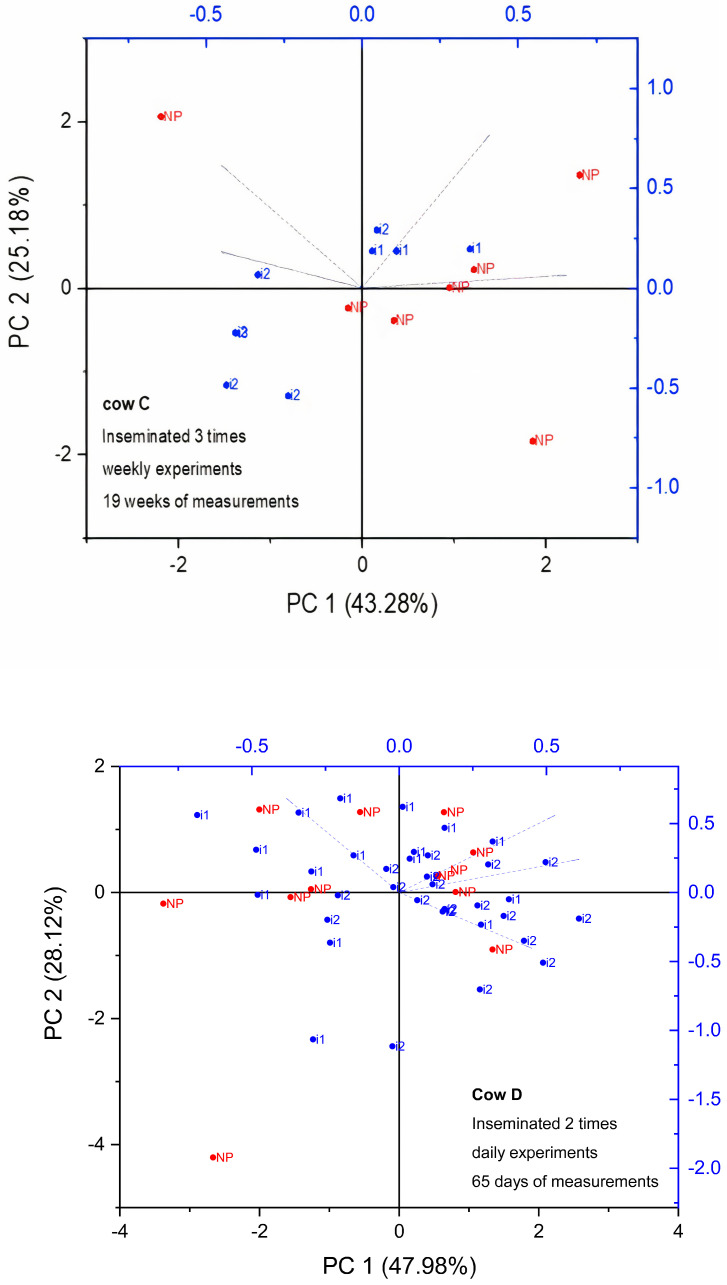
PCA biplot of the data corresponding to non-pregnant cows. The non-pregnant state has been marked as red NP” and the inseminated state as blue i. Cow C was measured weekly for 19 weeks, while cow D was measured daily for 65 days. The number of inseminations is presented as the number 1, 2 or 3 standing beside the letter i.

**Figure 8 sensors-24-02742-f008:**
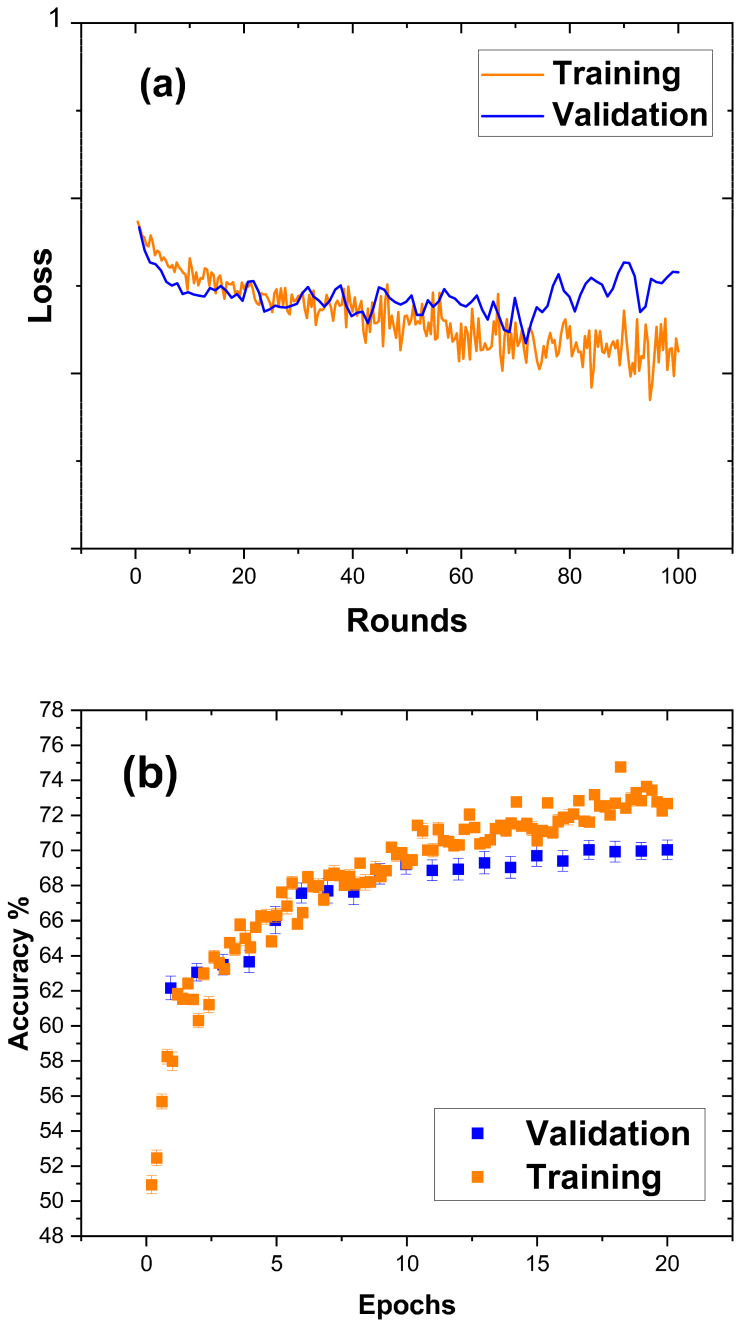
Generated learning curves from ANNs trained with an equal number of microwave dielectric milk measurements from pregnant and nonpregnant cows. (**a**) Loss plot, (**b**) accuracy plot.

**Table 1 sensors-24-02742-t001:** Pearson correlation matrix between CC fitting and physiological parameters for seven pregnant cows.

	Δε	τ	α	$\sigma_{dc}$
weather temperature	0.11	0.21	−0.21	0.26
*p*-value	0.03	0.002	0.002	<0.001
Fat %	** −0.66 **	−0.01	−0.08	−0.34
*p*-value	** <0.001 **	0.004	0.49	0.015
Density %	0.16	0.19	0.05	−0.04
*p*-value	0.95	0.99	0.036	0.91
Lactose %	−0.21	0.21	0.004	−0.26
*p*-value	<0.001	0.05	0.06	0.08
SNF (solids-not-fat) %	−0.23	0.21	0.002	−0.25
*p*-value	0.01	0.03	0.16	0.01
Protein%	−0.18	0.11	−0.03	−0.18
*p*-value	0.015	0.005	0.003	0.008

## Data Availability

The original contributions presented in the study are included in the article/supplementary material, further inquiries can be directed to the corresponding author/s.

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
