# Peer review of "Microwave Dielectric Response of Bovine Milk as Pregnancy Detection Tool in Dairy Cows"

_sensors, 2024, doi:10.3390/s24092742_

Round 1

Reviewer 1 Report

Comments and Suggestions for Authors

I find the paper to be interesting in terms of novelty and content, especially the use of Microwave sensors to monitor bovine milk. However, I feel that the authors should improve the writing of the paper as there are several grammatical errors in the manuscript. The comments are as follow:

1. The microwave sensors used in the study should be presented.

2. The prediction of pregnancy is performed using PCA and ANN. a comparison with other machine learning algorithms such as SVM and random forests is recommended.

3. The figure should have high quality.

4. There are fewer references, and the discussion of the latest work about phase spcae graph convolutional network in ieee tii can be appropriately added.

5. Figure 6 shows that the model has not reached a stable state after 20 epochs of training, and the prediction accuracy is also low. Why did the authors increase the number of training epochs. The interpretation of the error should be discussed in detail.

Author Response

Please see attachment. The tracked changes has been uploaded to non-published material

Reviewer 2 Report

Comments and Suggestions for Authors

Overall the study is very interesting and testing a novel concept. The objectives laid out in the beginning seem to have been achieved. One suggestion would be to include a statement as to how attainable it would be to implement this protocol on a farm versus methods that are used now. Ability to be tested on farm and intensity of sampling needed should be addressed.  

There is not a statement in the statistics section describing a power calculation or describing how the author's determined how many samples they needed to collect, can this be addressed how it was done, or if not needed please describe why.

Line 355 is there supposed to be one point labeled non pregnant that is blue?

I wonder if it would be interesting to select a a day for pregnancy in which samples were collected for various cows and match those with non-pregnant cows that were inseminated on a similar day and put all those cows into one PCA plot and see if there is clustering at the population level as well, not just as the cow level.

Comments on the Quality of English Language

Line 44: unhealthy and healthy?

Line 73: false negative and false positive

Author Response

Please see attachment.The tracked changes has been uploaded to non-published material

Reviewer 3 Report

Comments and Suggestions for Authors

Thank you for inviting me to review this interesting manuscript that describes a study investigating the potential for microwave dielectric response of milk as an aid to pregnancy diagnosis in dairy cows. This technique is not well described for this purpose therefore this work is novel and of potential interest to the bovine industry.  

A major concern I have with the design of this study is that the studied test (microwave dielectric response) has been compared to manual palpation as a method of diagnosing pregnancy.  The choice to use manual palpation rather than transrectal ultrasound as a diagnosis of pregnancy really needs to be discussed as manual palpation is no longer considered optimal for pregnancy diagnosis and is a limitation to this work.  There is also no discussion about the potential for error as a negative pregnancy diagnosis at 42 days post-AI is assumed to be a failure to conceive, and therefore also be negative at 15 days post-AI, whereas of course there is the possibility that cows did conceive and were positive at 15 days post-AI but have experienced embryo loss.  We cannot accurately diagnose pregnancy as early as 15 days so this will always be a limitation regardless of which 'gold standard' test is used for validation but this still requires discussion as a limitation of the study.     

Please see PDF for detailed comments.  

Comments on the Quality of English Language

The English in the manuscript is excellent, there are a couple of very minor grammatical errors - see detailed comments in the PDF.  

Author Response

(The authors gave the same response as above.)

Round 2

Reviewer 1 Report

Comments and Suggestions for Authors

Most problems has been revisied, The doI required by the authors is https://doi.org/10.1109/TII.2024.3363089; https://doi.org/10.1016/j.chaos.2023.114170

Author Response

We highly appreciate the recommendation of reviewer 1 regarding the  incorporation of two recent studies in the field of neural network algorithms:

The first study is available at https://doi.org/10.1109/TII.2024.3363089; The second study, is accesible via :  https://doi.org/10.1016/j.chaos.2023.114170   These studies delve into the development of a novel method named Phase Space Graph Convolutional Network (PSGCN) that allows identification of dynamic features underlying the signals in a chaotic system experiment.   We acknowledge the significant importance of this research. However, as outlined in our manuscript, our algorithm remains at a preliminary stage. It was developed using relatively small dataset, with no consideration given to temporal aspects. We focused on the classification of the dielectric data as either pregnant or non pregnant and built the deep neural network (SNN) capable of capturing complex patterns and relationships within the input data.    We believe that in future works, the increase of the amount of data  will allow us to conduct more comprehensive studies and accurately identify changes in the dynamics of our parameters. Subsequently, we will be well-positioned to apply advanced algorithms such the suggested by the reviewer.